# Deep Learning to Decipher the Progression and Morphology of Axonal Degeneration

**DOI:** 10.3390/cells10102539

**Published:** 2021-09-25

**Authors:** Alex Palumbo, Philipp Grüning, Svenja Kim Landt, Lara Eleen Heckmann, Luisa Bartram, Alessa Pabst, Charlotte Flory, Maulana Ikhsan, Sören Pietsch, Reinhard Schulz, Christopher Kren, Norbert Koop, Johannes Boltze, Amir Madany Mamlouk, Marietta Zille

**Affiliations:** 1Fraunhofer Research and Development Center for Marine and Cellular Biotechnology EMB, 23562 Lübeck, Germany; alex.palumbo@uni-luebeck.de (A.P.); svenja-kim.landt@gmx.net (S.K.L.); LaraEleen.Heckmann@kgu.de (L.E.H.); luisa.bartram@web.de (L.B.); apabst@ukaachen.de (A.P.); charlotte.flory@leibniz-hpi.de (C.F.); maulanaikhsan@unimal.ac.id (M.I.); soeren.pietsch@medizin.uni-leipzig.de (S.P.); johannes.boltze@warwick.ac.uk (J.B.); 2Institute for Medical and Marine Biotechnology, University of Lübeck, 23562 Lübeck, Germany; 3Institute for Experimental and Clinical Pharmacology and Toxicology, University of Lübeck, 23562 Lübeck, Germany; 4Institute for Neuro- and Bioinformatics, University of Lübeck, 23562 Lübeck, Germany; gruening@inb.uni-luebeck.de (P.G.); madany@inb.uni-luebeck.de (A.M.M.); 5Faculty of Medicine, Malikussaleh University, Lhokseumawe 24355, Indonesia; 6Department of Neonatology, Universitätsklinikum Leipzig, 04103 Leipzig, Germany; 7Wissenschaftliche Werkstätten, University of Lübeck, 23562 Lübeck, Germany; reinhard.schulz@uni-luebeck.de; 8Medical Laser Center Lübeck GmbH, 23562 Lübeck, Germany; christopher.kren@uni-luebeck.de (C.K.); n.koop@uni-luebeck.de (N.K.); 9School of Life Sciences, The University of Warwick, Gibbet Hill Campus, Coventry CV4 7AL, UK; 10Department of Pharmaceutical Sciences, Division of Pharmacology and Toxicology, University of Vienna, 1090 Vienna, Austria

**Keywords:** axon, brain hemorrhage, cortical neurons, cell culture, machine learning, microfluidic, microscopy, stroke, time-lapse

## Abstract

Axonal degeneration (AxD) is a pathological hallmark of many neurodegenerative diseases. Deciphering the morphological patterns of AxD will help to understand the underlying mechanisms and develop effective therapies. Here, we evaluated the progression of AxD in cortical neurons using a novel microfluidic device together with a deep learning tool that we developed for the enhanced-throughput analysis of AxD on microscopic images. The trained convolutional neural network (CNN) sensitively and specifically segmented the features of AxD including axons, axonal swellings, and axonal fragments. Its performance exceeded that of the human evaluators. In an in vitro model of AxD in hemorrhagic stroke induced by the hemolysis product hemin, we detected a time-dependent degeneration of axons leading to a decrease in axon area, while axonal swelling and fragment areas increased. Axonal swellings preceded axon fragmentation, suggesting that swellings may be reliable predictors of AxD. Using a recurrent neural network (RNN), we identified four morphological patterns of AxD (granular, retraction, swelling, and transport degeneration). These findings indicate a morphological heterogeneity of AxD in hemorrhagic stroke. Our EntireAxon platform enables the systematic analysis of axons and AxD in time-lapse microscopy and unravels a so-far unknown intricacy in which AxD can occur in a disease context.

## 1. Introduction

Axonal degeneration (AxD) is a process in which axons disintegrate physiologically during nervous system development and aging, or as a pathological element of degenerative nervous system diseases [1,2,3]. Apart from axonal fragments, axon swellings (also called axonal beadings, bubblings or spheroids) are a hallmark of degenerating axons [2,4,5], containing disorganized cytoskeleton and organelles resulting from an interruption of axonal transport [6,7,8].

It is known that axons disintegrate in different ways depending on the biological context. During development and neural circuit assembly, inappropriately grown axons can undergo axonal retraction, axonal shedding, or local AxD [9,10]. Axonal retraction is characterized by retraction bulb formation at the distal tip and subsequent pullback [9]. During axonal shedding, the axon retracts, leaving behind small pieces of its distal part (axosomes) [11]. Local AxD is characterized by axon disintegration into separated axonal fragments [10]. Acutely and chronically injured axons may degenerate retrogradely (distal-to-proximal direction, dying-back), anterogradely (proximal-to-distal direction), or in a Wallerian degeneration pattern (distal part of the axon from injury site), ultimately resulting in the generation of axonal fragments [6,12,13]. However, AxD patterns have been described primarily in extracerebral axons in models of nutrient deprivation or axotomy.

Not much is known about AxD in cortical neurons subjected to a disease-specific cytotoxic micromilieu. A distinct pathological micromilieu has recently been observed for hemorrhagic stroke, after which the lysis of erythrocytes from the hematoma leads to the release of the cytotoxic product hemin [14,15]. Patients suffering from hemorrhagic stroke often experience AxD that is associated with worse motor and functional outcome [16,17]. Importantly, AxD occurs in the subacute stages of hemorrhagic stroke. Thus, addressing AxD may not only provide a new therapeutic target, but also a much wider time window for intervention. Since not much is known about the mechanisms, morphological patterns, and the temporal progression of AxD in the context of hemorrhagic stroke, we here sought to examine the progression of AxD and its associated morphological alterations.

Several cell culture systems, such as Campenot chambers and microfluidic devices, have been developed to spatially separate axons from their somata, allowing researchers to study AxD at a molecular level and to dissect the axon-soma relationship in AxD [18,19]. Whereas Campenot chambers facilitate the collection of axonal material due to their open structure [20], they are only suitable for neurons that project axons robust enough to cross the vacuum–grease barrier needed to affix the Teflon piece that separates the somata from their axons. This makes Campenot chambers unsuitable for most central nervous system neurons [21]. In microfluidic devices, the spatial separation is enabled by a microflux established by a medium volume difference between the two opposing compartments. These opposing compartments are separated by microgrooves through which axons grow [22]. The major limiting factor of commercially available microfluidic devices to study AxD is that they are single, individual systems. Hence, they can only be used to assess one condition, which is time-consuming and precludes high-throughput analyses [19,23].

As the disintegration of the axons endures from minutes to hours [13,24], it is necessary to continuously monitor the spatiotemporal progression of AxD and its morphological hallmarks. On the one hand, axonal fragments have been identified by binarizing microscopic images [25,26]. Structures that are continuously connected are interpreted as axons, while isolated, non-connected elongated structures are categorized as axonal fragments. The ratio of axonal fragments to total axons is indicated by the so-called AxD index [25,26]. On the other hand, the occurrence and motility of axonal swellings have been determined previously by defining a region of interest, kymograph analysis, or calculating the ratio of the number of swellings compared to the axon length [8,27,28,29].

However, conventional software solutions fail to automatically detect and quantify high axon numbers as well as axonal swellings and fragments in phase-contrast microscopic images. The reason may be two-fold: (1) binarization can lead to information loss and low sensitivity, as thin axons may not be recognized; and (2) The analysis requires subjective and time-consuming manual annotations, e.g., thresholding and defining the region of interest [27,30,31]. So far, immunostained images have been used to investigate the morphological changes in AxD, as the analysis of phase-contrast images is limited by the lower target-to-background signal. Immunofluorescence images, however, entail certain disadvantages, such as photobleaching and the requirement for cell fixation, which restricts observations to a single time point. Thus, a software tool for the automatized detection and quantification of the morphological patterns of AxD in long-term live cell imaging is required to improve both sensitivity and throughput to overcome the current limitations in understanding AxD.

In this study, we demonstrate that cortical axons undergo AxD after the exposure to the hemolysis product hemin, with axonal swellings preceding axon fragmentation. Deep learning further detected the occurrence of four AxD patterns characterized as granular, retraction, swelling, and transport degeneration. This may inform downstream AxD and neurodegeneration research in health and disease. We also provide tools for the enhanced throughput analysis of AxD, including a microfluidic device containing 16 independent experimental units and the deep learning platform “EntireAxon” to analyze AxD, which will help augment our understanding of AxD and may also support the development of novel treatment approaches for neurodegenerative diseases.

## 2. Materials and Methods

Chemicals and reagents are listed in Appendix A.

### 2.1. Fabrication of an Enhanced Throughput Microfluidic Device Based on Soft Lithographic Replica Molding

In total, 32 wells were milled in a polymethyl methacrylate (PMMA) plate of the size of a conventional cell culture plate using a universal milling machine (Mikron WF21C, Mikron Holding AG) with a 1 mm triple tooth cutter (HSS-CO8 Type N, Holex, Munich, Germany) at a precision of 0.01 mm. During the milling procedure, we applied a half-synthetic cooling lubricant (Opta Cool 600 HS, Fuchs Wisura GmbH, Bremen, Germany) on a mineral base to reduce the debris. In addition, we milled screw holes in the intermediate spaces between each microfluidic unit to later detach the PMMA from the negative casting mold. To remove debris, we washed the PMMA plate by sonication (Sonicator Elmasonic S, Elma Schmidbauer GmbH, Singen, Germany) at room temperature for 30 min. Next, we lasered the microgrooves on the PMMA plate to connect both milled compartments of each individual microfluidic unit using an Excimerlaser (Excistar XS 193 nm, Coherent, Santa Clara, CA, USA). The PMMA plate was then washed again by sonication at room temperature for 30 min.

Polydimethylsiloxane (PDMS) was prepared in a 1:10 ratio and mixed properly before inducing vacuum at 0.5 Torr in a vacuum desiccator (VDC-31, Jeio Tech, Daejeon, Korea) for 30 min. After the PDMS was poured into an empty aluminum basin to cover the ground, we applied the vacuum at 0.5 Torr for 30 min to remove the air bubbles. The PDMS was cured at room temperature for 48 h. We put the PMMA plate on top of the PDMS ground with the milled and lasered structures showing upward. Half of each well of the microfluidic units was filled with PDMS before curing at room temperature for 48 h. We mixed epoxy solution in a 1:1 ratio and poured it over the microfluidic device to cover its surface by at least 1 cm. Vacuum was applied at 0.5 Torr for 10 min to remove all air bubbles located above the channel side of the microfluidic device. The epoxy was cured at room temperature for a minimum of 2 h. We subsequently detached the epoxy from the PMMA plate via a metallic block that consisted of screw holes in the intermediate spaces between the individual systems. The epoxy represented a negative casting mold to produce the microfluidic devices using PDMS.

PDMS was prepared as described above. We poured the PDMS into the negative epoxy casting mold and applied vacuum at 0.5 Torr for 30 min. The liquid PDMS was cured at 75 °C for 2 h to induce polymerization. We peeled the microfluidic devices from the casting mold and punched the wells with an 8 mm biopsy punch (DocCheck Shop GmbH, Cologne, Germany) to ensure a sufficient amount of medium for cell culture. We cleaned customized 115 × 78 × 1 mm glass slides by sonication (Sonicator Elmasonic S, Elma Schmidbauer GmbH) and subsequently cleaned them by ethanol before plasma treatment (High Power Expanded Plasma Cleaner, Harrick Plasma, Ithaca, NY, USA). Plasma was applied at 45 W and 0.5 Torr for 2 min to activate the silanol groups of the glass slides and the microfluidic devices, enabling firm attachment.

We washed the microfluidic devices with ethanol and then twice with distilled water to remove any debris. After aspirating the distilled water, except from the inside of the compartments, 0.1 mg/mL of poly-d-lysine solution in 0.02 M borate buffer (0.25% (*w*/*v*) borate acid, 0.38% (*w*/*v*) sodium tetraborate in distilled water, pH 8.5) was used for coating at 4 °C overnight. We aspirated the poly-d-lysine the next morning, not removing it from the compartments, and added 50 µg/mL of laminin as a second coating surface for incubation at 4 °C overnight. On the day of neuron isolation, the microfluidic devices were washed twice with pre-warmed medium after aspirating the laminin. Immediately prior to cell seeding, we aspirated the medium from the wells without removing it from the compartments.

### 2.2. Experimental Animals

Crl:CD1 (ICR) Swiss outbred mice (Charles River) were used. The animals were kept at 20–22 °C and 30–70% humidity in a 12-h/12-h light/dark cycle, and were fed a standard chow diet (Altromin Spezialfutter GmbH, Lage, Germany) ad libitum. Animal experiments were performed in accordance with the German Animal Welfare Act and the corresponding regulations and were approved by the local animal ethics committee (Ministerium für Landwirtschaft, Umwelt und ländliche Räume, Kiel, Germany, under the prospective contingent animal license number 2017-07-06 Zille).

### 2.3. Isolation and Culture of Primary Cortical Neurons

We isolated the primary cortical neurons from the murine E14 embryos after decapitation as previously described [15]. We seeded the neurons at a density of 10,000 cells/mm^2^ in 5 µL MEM + Glutamax medium into one compartment (soma compartment) of each microfluidic unit of the device. The cells were allowed to adhere at 37 °C for 30 min. To promote directional axon growth into the other compartment (axonal compartment) by medium microflux, 150 µL of MEM + Glutamax medium were applied to the well of the soma compartment, while 100 µL were added to the well of the axonal compartment. Neurons were cultured at 37 °C in a humidified 5% CO_2_ atmosphere. The next day, we changed from MEM + Glutamax medium to Neurobasal Plus Medium containing 2% B-27 Plus Supplement, 1 mM sodium pyruvate and 1% penicillin/streptomycin. The volume differences among the wells ensured the microflux for the directional axonal growth over the following days.

### 2.4. Immunofluorescence

Soma and axonal compartments in the microfluidic units were fixed at room temperature for 1 h in 4% formaldehyde solution in phosphate-buffered saline (PBS). They were washed twice with PBS and permeabilized with blocking solution (2% BSA, 0.5% Triton-X-100 and 1× PBS) at room temperature for 1 h. We incubated the neurons/axons on both compartments with primary antibodies against synaptophysin (1:250) and MAP2 (1:4000) at 4 °C overnight. The next day, both compartments were washed three times with PBS and incubated with the secondary antibodies anti-mouse Alexa Fluor 546 (1:500) and anti-rabbit Alexa Fluor 488 (1:500) at room temperature for 1 h. After washing three times with PBS, both compartments were incubated with DAPI (1 µg/mL) for nuclear counterstaining at room temperature for 10 min. Both compartments were washed again three times with PBS prior to fluorescence microscopy. An Olympus IX81 time-lapse microscope (Olympus Deutschland GmbH, Hamburg, Germany) with a 10× objective (0.3 NA Ph1) and camera F-View soft Imaging system was used at room temperature. Images were acquired with Cell^TM^ software (Olympus Deutschland GmbH) and further processed via ImageJ (see Section 2.10).

### 2.5. Selection of Microfluidic Units for Hemin Treatment and Time-Lapse Recording

After 6 or 7 days in culture, microfluidic devices were considered for recording if they met the following inclusion criteria: (i) axon growth through at least 80% of all microgrooves, and (ii) axon length of at least 150 µm from the end of the microgrooves. All included microfluidic units were randomly assigned to the experimental conditions.

### 2.6. Time-Lapse Recording of Axonal Degeneration

Axons were treated with 0 (vehicle), 50, 100, and 200 µM hemin. For the treatment, the medium was removed from the wells of the microfluidic units. Then, hemin was diluted in the collected media and added back to the respective wells. The media volume between the two wells was equalized during the treatment to prevent any microflux. All microfluidic units were recorded immediately after each other. We started the recordings at 1 h after treatment to allow for the adjustment of the well plates to the humidity of the incubation chamber of the microscope and the setup of the recording positions. We recorded AxD in Neurobasal Plus Medium containing 2% B-27 Plus Supplement, 1 mM sodium pyruvate, and 1% penicillin/streptomycin with a 30-min interval for 24 h using an Olympus IX81 time-lapse microscope (see Section 2.4) at 37 °C, 5% CO_2_, and 65% humidity.

### 2.7. Live Cell Fluorescent Staining

To evaluate axonal vitality, we washed the axonal compartment once with PBS and incubated the axonal compartment with calcein AM (4 µM) in PBS for 30 min at 37 °C at the end of the time-lapse recording or in 4 h intervals upon hemin treatment. An Olympus IX81 time-lapse microscope (see Section 2.4) was used to record the respective images at 37 °C, 5% CO_2_, and 65% humidity.

### 2.8. Training of the EntireAxon CNN for the Segmentation of Phase-Contrast Microscopic Images

We trained the EntireAxon CNN for the image-wise semantic segmentation of AxD features in a supervised manner. To this end, we adapted a standard u-net with ResNet-50 encoder [32] to automatically determine the class probability for each pixel of an input image. Our segmentation aimed to classify each pixel of a microscopic image of a time-lapse recording into one of four classes: ‘background’, ‘axon’, ‘axonal swelling’, and ‘axonal fragment’.

For the training dataset, we selected 33 images and created corresponding image labels (masks) using GIMP software (v.2.10.14, RRID:SCR_003182). For each image, a label image with the same height and width was created, in which each pixel value denotes a pixel class. Specifically, the classes ‘background’, ‘axon’, ‘axonal swelling’ and ‘axonal fragment’ had the values 0, 1, 2, and 3, respectively. For each pixel of the input image, we retained 4 values that reflect the probability distribution of the pixel over the 4 classes. We assigned each pixel the most probable class to create a segmentation map. During training, the CNN observed an input image, produced an output, and compared this output to the label. The weights of the network were adapted via backpropagation so that the output better fitted the label. The weight changes were derived from a pixelwise loss function, i.e., the cross-entropy loss:(1)$$\mathrm{Loss}\left(P,\;Y\right)=-\underset{x,y,c}{\sum }Y\left(x,y,c\right)\;\log\left(P\left(x,y,c\right)\right)$$
with P(x,y,c) and Y(x,y,c) being the probability of class c at pixel (x,y) for the prediction and ground truth of the network, respectively.

We trained a mean ensemble consisting of 8 neural networks for 180 epochs using the Adam optimizer, a batch size of 4, and a learning rate of 0.001 that decreased by a factor of 10 after every 60 epochs. The input images were standardized by the image-net mean and standard deviation [28]. For data augmentation, we used random cropping (size 512 × 512), image flipping along the horizontal axis, and rotation by a random angle between −90° and +90°.

### 2.9. Validation of the EntireAxon CNN Compared to Human Evaluatorts

To measure how well the EntireAxon CNN segments unknown images, we used a second validation set comprising eight images that were labeled by three human evaluators (A.P. (Alex Palumbo), S.K.L., L.E.H.). Importantly, the EntireAxon CNN did not update its parameters during training to fit the validation set, but only used the training set.

For each image, the EntireAxon CNN inferred a segmentation. We generated a binary mask from the prediction of the network, where 1 denotes the respective class and 0 all other classes. We computed a binary label mask in the same manner. We counted the true positive (TP), false positive (FP), and false negative (FN) pixels and computed the recall (sensitivity) and precision [33]:(2)$$\mathrm{Recall}=\frac{\mathrm{TP}}{\mathrm{TP}+\mathrm{FN}}$$
(3)$$\mathrm{Precision}=\frac{\mathrm{TP}}{\mathrm{TP}+\mathrm{FP}}$$

Recall and precision were calculated separately for each class on each validation image. The mean recall and precision over all eight validation images were determined subsequently.

A mean of 96.42% of pixels in the axonal images were ‘background’ pixels, while only 2.77% represented the class ‘axon’, 0.58% ‘axonal swelling’, and 0.23% ‘axonal fragment’ pixels. This reflects a challenging degree of class imbalance, where the probability of having any positives for a class in a validation image is low. Thus, we did not use the computed recall and precision of the individual images or the mean recall and precision to compute the mean F1 score, i.e., the harmonic mean of recall and precision. This has been shown to lead to bias, especially when a high degree of class imbalance is present in the dataset [33], as it may result in undefined values for an image for recall (due to the absence of TP), precision (in case the CNN does not recognize the few positives), and F1 score (in case either recall or precision are undefined). To avoid bias, we computed the total TP, FP, and FN of all validation images from which we calculated the mean F1 score [33]:(4)$$\mathrm{mean}\;F1\;\mathrm{score}=\frac{2\times {\mathrm{TP}}_{\mathrm{total}}}{2\times {\mathrm{TP}}_{\mathrm{total}\;}+{\;\mathrm{FP}}_{\mathrm{total}}+{\;\mathrm{FN}}_{\mathrm{total}}}$$

In addition, we computed a consensus label between human evaluator 1 and 2, between 1 and 3, and between 2 and 3, and compared the EntireAxon CNN versus the remaining evaluator (human evaluator 3, 2, and 1, respectively) to the consensus labels. Mean F1 scores for all classes were computed as described above.

### 2.10. Image Preprocessing

Prior to the analysis of AxD after hemin exposure, we preprocessed the time-lapse recordings in ImageJ (v1.52a, RRID: RRID:SCR_003070) using a custom-written macro. Specifically, each individual recording was converted from a 16-bit into an 8-bit recording to make it compatible with the ImageNet (8-bit) pre-trained ResNet-50. The recording was aligned automatically with the ImageJ plug-in “Linear Stack Alignment with SIFT” as described previously [34]. The following settings were used: initial Gaussian blur of 1.6 pixels, 3 steps per scale octave, minimum image size of 64 pixels, maximum image size of 1024 pixels, feature descriptor size of 4, 8 feature descriptor orientation bins, closest/next closest ratio of 0.92, maximal alignment error of 25 pixels, inlier ratio of 0.05, expected transformation as rigid, “interpolate” and “show info” checked. Black edges that appeared on the recording after alignment were cropped.

### 2.11. AxD Analysis Using the EntireAxon CNN

All recordings of AxD after hemin exposure were automatically analyzed by the trained EntireAxon CNN, which classified each pixel as one of the 4 different classes: ‘background’, ‘axon’, ‘axonal swelling’, or ‘axonal fragment’. For each experimental condition (i.e., hemin concentration), the sum percentage of all pixels per class on all images of that experimental day were added at each timepoint (Axon_t1.5–24h_, Axonal swelling_t1.5–24h_, Axonal fragment_t1.5–24h_). To determine the changes for the classes ‘axon’, ‘axonal swelling’, and ‘axonal fragment’ over time, we calculated the sum percentage of pixels for all given time points (t_i_ with I = 1.5 to 24 h) of the corresponding class over the sum of the pixels of all three classes at baseline:(5)$$\mathrm{normalized}{}^{‘}\mathrm{class}\;{}^{’}\;\mathrm{area}\;\left(t_{i}\right)=\frac{{{}^{‘}\mathrm{Class}{}^{’}}_{\mathrm{ti}}}{{\mathrm{Axon}}_{t1.5h}+{\mathrm{Axonalswelling}}_{t1.5h}+{\mathrm{Axonalfragments}}_{t1.5h}}\times 100$$

The area under the curve was calculated as the cumulative measurement of the effect of hemin on axonal degeneration over 24 h as follows:(6)$$\mathrm{AUC}=\sum (\left({{}^{‘}\mathrm{Class}{}^{’}}_{\mathrm{ti}}+{{}^{‘}\mathrm{Class}{}^{’}}_{\mathrm{ti}+1}\right)\times \left(t_{i+1}-t_{i}\right))$$

### 2.12. Classification of the Morphological Patterns of AxD Using an Attention-Based RNN

We used the segmentation videos derived from the original microscopic images, using the RNN to identify 4 morphological patterns of AxD: granular, retraction, swelling, and transport degeneration. We manually annotated the segmentation videos of individual, degenerating axons to classify each AxD pattern. To reduce the dimensions of the input, the segmentation video was converted into a series of normalized histograms (*H*), one for each (time) frame. Thus, the RNN did not operate on the microscopic images directly, but rather on more efficient representations of the data. To compute a histogram for a frame t_i_, we compared the pixels of the frames t_i_ and t_i+1_. Each pixel was assigned to 1 of 16 classes that consisted of pairs $\left(c_{1},c_{2}\right)\in \;{\left\{0,1,2,3\right\}}^{2}$ of the 4 segmentation classes (i.e., 4 times 4 possible configurations, 16 class pairs). For example, the class (background, axon) means that in frame t_i_, the pixel was classified as background, while in frame t_i+1_, it was an axon pixel. Therefore, for T time steps, we computed T-1 histograms. $H_{0}\left(t_{i},\;\left(c_{1},c_{2}\right)\right)$ is the number of pixels that belong to class $c_{1}$ at time-frame $t_{i}$ and that belong to $c_{2}$ at timeframe $t_{i+1}$. In addition, we normalized each histogram to sum up to 1 (i.e., we divided by the sum over all pairs):(7)$$H\left(t_{i},\;\left(c_{1},c_{2}\right)\right)={\;H}_{0}\left(t_{i},\;\left(c_{1},c_{2}\right)\right)\;/\;\underset{a,b}{\sum }H_{0}\left(t_{i},\;\left(a,b\right)\right)$$

Of note, the histograms were computed over small patches (height and width < 90 pixels) during training and during inference on windows of size 32 × 32 pixels.

We used an encoder-decoder RNN with attention [35]. The encoder fenc consisted of a gated recurrent unit (GRU) that obtained the histogram time sequence H as input. The encoder computed the hidden representation of the histograms:(8)V = fenc(H); V∈ℝT × d, H∈ℝT × 16

For our purpose, we used an architecture that was able to base the decision for a degeneration class on the previous class predictions. To this end, the output o→i was computed iteratively in C + 1 steps as a sum of the previous output and the output of the decoder fdec:(9)o→i=o→i−1+fdec(𝓸(o→i−1),s→i−1,V);o→∈ℝC,s→i−1∈ℝd
(10)fdec(𝓸(o→i−1),s→i−1,V)=Woutz→i; Wout∈ℝC × d

C is the number of degeneration classes (4), d is the hidden dimension (we used 256), and i=1,...,C+1. 𝓸 is the sigmoid function. The decoder employed a GRU that depended on the context vector c→i and the hidden state vector s→i−1:(11)z→i,s→i= GRU(c→i,s→i−1);z→i∈ℝd

The entries of the initial hidden vector s→0 and output vector o→0 were all zero. The context vector is a weighted sum of the encoder representations. At each iteration, these weights can change, enabling the network to focus on different time steps. We assumed that a specific pattern of degeneration happened only in a limited number of timeframes that was lower than the whole input video. The weights depended on the current state of the decoder and the current output:(12)c→i= VTα→i; α→i ∈ℝT
(13)α→i= Softmax( Watt [s→i−1, ReLU(Wino→i−1)]); Watt∈ℝT × 2d, Win∈ℝd × C

Here, $\left[\vec{a},\;\vec{b}\right]$ is the concatenation of two vectors. The final output y is normalized by the sigmoid function:(14)$$y\;=𝓸\left({\vec{o}}_{C+1\;}\right)\;\in \;{\left(0,\;1\right)}^{C}.$$

Apart from the weights used by the GRUs, $W_{in},\;W_{att},$ and $W_{out}$ are learnable weights.

The EntireAxon RNN was trained with 162 images for 60 epochs using the lamb optimizer [36] with a batch size of 128. We used a learning rate of 0.01 that was reduced by a factor of 10 every 15 epochs and an additional weight decay of 0.0001. The two GRUs (encoder and decoder) contained three layers, and we used dropout with a *p*-value of 0.9. To increase the RNN robustness against varying axon thickness, we also added eroded versions of the segmentation data using a cross-shape as kernel with the sizes three, five, and seven. Accordingly, each image existed six times in the dataset, with three eroded versions and three unchanged copies, to keep a 50% chance of having the original image for training.

### 2.13. RNN Cluster Analysis

The unnormalized class output ${\vec{o}}_{C+1}$ was computed by the matrix-vector product $W_{out}{\vec{z}}_{C+1}$ where ${\vec{z}}_{C+1}$ was a 256-dimensional vector representation of the input sample, computed by the model. For the classes to be linearly separable, the vector representations of each class needed to be close to each other in the 256-dimensional space. To visualize the relationships of the specific samples, we employed t-distributed stochastic neighborhood embedding (T-SNE) to compute a 2-dimensional representation of the high-dimensional data.

### 2.14. Ten-Fold Cross-Validation of the RNN

To validate the RNN, we used 10-fold cross-validation [37]. The dataset S was divided into 10 subsets, ensuring that each subset included at least 1 sample of each class: $S=\cup_{i=1}^{10}S_{i};\;S_{i}\;\cap \;S_{j}=\emptyset ,\;i\neq j$. We trained 10 models for *i* = 1,...,10 on $Train_{i}=S\;/\;S_{i}$ and tested them on $Test_{i}=S_{i}$. Subsequently, we combined and evaluated all test samples $Test=\cup_{i=1}^{10}Test_{i}$. Mean recall, precision, and F1 score were determined as described above.

### 2.15. Analysis of the Morphological Pattern of AxD Using the EntireAxon RNN

After hemin exposure, all AxD segmentations were automatically analyzed with the trained EntireAxon RNN, which predicted the occurrence of the 4 morphological patterns of AxD in a pixel-wise manner. Of note, a pixel can be predicted to belong to 0, 1, or multiple morphological patterns. Only pixels previously identified as degenerated over time were considered by applying a ‘fragmentation mask’ that included all no-background pixels that changed to either background or fragment during the recording time.

For each experimental condition (i.e., hemin concentration), the percentage of the occurrence of each morphological pattern was calculated as the sum of all pixels per morphological pattern on all images of that experimental day divided by the ‘fragmentation mask’ as follows:(15)$${}^{‘}\mathrm{morphological}\;\mathrm{pattern}{}^{’}\left[\%\right]=\frac{\sum {\mathrm{pixel}\;\mathrm{of}\;\mathrm{morphological}\;\mathrm{pattern}}_{i}}{\sum \mathrm{pixel}\;\mathrm{no}\;\mathrm{background}\to \mathrm{background}\;\mathrm{or}\;\mathrm{fragment}}\times 100$$

### 2.16. Statistical Analysis

Six biological replicates for each concentration were employed in each experiment to assess the hemin-induced AxD. We did not perform an a priori power analysis, as this was an exploratory study. Normality was evaluated with the Kolmogorov–Smirnov test, variance homogeneity using the Levené test, and sphericity by the Mauchly test. When the data were normally distributed and variance homogeneity was met, one-way ANOVA was performed, followed by the Bonferroni post-hoc test. In the case that the data were not normally distributed, the Kruskal–Wallis test was performed for multiple comparisons of independent groups, followed by the post-hoc Mann–Whitney U test with α-correction according to Bonferroni-Holm to adjust for the inflation of type I error due to multiple testing. For repeated testing with covariates, a repeated measures ANOVA was performed with Greenhouse–Geisser adjustment if the sphericity was not given. Linear regressions were performed for AxD patterns. Data are represented as mean ± 95% confidence interval (CI) except for the nonparametric data of the AUC for axonal fragments, where medians are given. A value of *p* < 0.05 was considered statistically significant. The detailed statistical analyses can be found in Appendix A. All statistical analyses were performed with IBM SPSS version 23 (RRID:SCR_002865), except linear regressions that were performed with GraphPad Prism version 8 (RRID:SCR_002798).

## 3. Results

### 3.1. The EntireAxon Platform for the Longitudinal Study of Axonal Degeneration

To enable the systematic analysis of AxD in vitro, we manufactured a microfluidic device containing 16 individual microfluidic units (Figure 1 and Appendix A) that can be investigated in parallel and recorded simultaneously, which reduces the manufacturing, treatment, and recording time. We demonstrate that this novel device allows the spatial separation of somata (microtubule-associated protein 2-positive) from their axons (synaptophysin-positive), which is leakproof, as somata were not detected on the axonal side (Figure 1d).

Then, we trained a CNN, the EntireAxon, to segment all relevant features of AxD, i.e., axons, axonal swellings, and axonal fragments (Figure 2). Whereas the EntireAxon CNN recognized the class ‘background’ better than the three axon classes ‘axon’, ‘axonal swelling’, and ‘axonal fragment’ (mean F1 score: 0.995), axon-specific segmentation revealed the highest mean F1 score for the class ‘axon’ (0.780), followed by the classes ‘axonal swelling’ (0.567), and ‘axonal fragment’ (0.301) (Table 1).

Next, we compared the performance of the EntireAxon CNN on the ground truth (human evaluator 1) with two additional human evaluators. The EntireAxon CNN reached higher mean F1 scores for all classes, except for the class ‘axonal fragment’, where human evaluator 2 outperformed the EntireAxon CNN (Table 2).

This may have been due to the fact that the EntireAxon CNN was trained on images labeled by the same human evaluator (1) that labeled the ground truth. To assess whether its performance is more generalizable across the different evaluators, we compared the EntireAxon CNN to each of the human evaluators on the consensus labels of the two other human evaluators (Figure 3 and Table 3). Visual inspection of the labels showed a wide overlap between the different evaluators. However, there was considerable uncertainty, especially for the classification of axonal fragments (Figure 3). When comparing the mean F1 scores for all classes, the EntireAxon reached similar or even higher scores than the other three evaluators (Table 3). This may be because pixels that were differentially assigned by the human evaluator, i.e., more difficult to classify, were excluded from the comparison.

Collectively, this suggests that the EntireAxon CNN sensitively and specifically recognizes axons and the morphological features of AxD.

### 3.2. Axonal Integrity Is Lost over Time with Axonal Swellings Preceding Axon Fragmentation

Conventional in vitro models of AxD rely on nutrient deprivation or axotomy and focus on the axons outside of the brain. However, AxD is not only an active and commonly observed process in the brain, but it is also believed to be caused by more complex mechanisms given the different microenvironments in which it may occur. A distinct pathological micromilieu has recently been observed for hemorrhagic stroke, after which the lysis of erythrocytes from the hematoma leads to the release of the cytotoxic product hemin [14,15]. Patients suffering from hemorrhagic stroke often experience AxD that is associated with worse motor and functional outcome [16,17]. Importantly, AxD occurs in the subacute stages of hemorrhagic stroke. Thus, addressing AxD may not only provide a new therapeutic target, but also a much wider time window for intervention. We therefore modelled hemorrhagic stroke by exposing axons from primary cortical neurons to the hemolysis product hemin, a commonly used agent to mimic hemorrhagic stroke in vitro [14,15,38], and investigated the progression of AxD over 24 h.

Hemin induced time-dependent morphological changes, leading to AxD compared to vehicle-treated axons (Figure 4 and Appendix A). Area-under-the-curve (AUC) analyses over 24 h revealed a significant decrease in the axon area in all three hemin concentrations (50 µM vs. 0 µM: *P* = 0.026; 100 µM vs. 0 µM: *P* = 0.018, 200 µM vs. 0 µM: *P* < 0.001). The axonal swelling area also increased in all three concentrations (50 µM vs. 0 µM: *P* = 0.012, 100 µM vs. 0 µM: *P* = 0.005, 200 µM vs. 0 µM: *P* = 0.016), while the axonal fragment area was elevated only for axons treated with 100 and 200 µM hemin (vs. 0 µM: *P* = 0.008, Figure 4c and Appendix A).

Comparing the time course of AxD between hemin- and vehicle-treated axons (0 µM), the axon area decreased starting at 11.5 h at 200 μM (*P* = 0.020, from 15 h *P* < 0.001), at 14 h at 100 μM (*P* = 0.040, from 18.5 h *P* < 0.001), and at 15 h at 50 μM (*P* = 0.018, from 19 h *P* < 0.001). Hemin treatment also elevated the axonal fragment area starting at 9 h at 200 μM (*P* = 0.037) and at 17 h at 100 μM hemin (*P* = 0.044). Interestingly, the axonal swelling area increased prior to the changes in axon and axonal fragment area, i.e., starting at 6 h at 200 µM (*P* = 0.010) and 100 µM (*P* = 0.019), and at 8 h at 50 µM hemin (*P* = 0.030). For the highest hemin concentration, the increase was only transient (until 18.5 h), suggesting that axonal swellings preceded the axon fragmentation (Appendix A), which can also be seen in the time-lapse recordings (Appendix A).

The results of the time course analysis were further substantiated by live cell fluorescent staining (calcein AM), which indicated the starting point of AxD after hemin treatment was between 8 and 12 h for 200 μM hemin, between 12 and 16 h for 100 μM hemin, and 16 and 20 h for 50 μM hemin (Appendix A). Taken together, AxD progression depends on the severity of the insult, and axonal swellings may be reliable predictors of AxD.

### 3.3. Deep Learning Deciphers Four Patterns of AxD

Axons disintegrate in different ways depending on the biological context. During development and neural circuit assembly, inappropriately grown axons can undergo axonal retraction, axonal shedding, or local AxD [9,10]. Axonal retraction is characterized by retraction bulb formation at the distal tip and subsequent pullback [9]. During axonal shedding, the axon retracts, leaving behind small pieces of its distal part (axosomes) [11]. Local AxD is characterized by axon disintegration into separated axonal fragments [10]. Acutely and chronically injured axons may degenerate retrogradely, anterogradely, or in a Wallerian degeneration pattern, ultimately resulting in the generation of axonal fragments [6,12,13]. Although organelles and other cargo are constantly transported anterogradely and retrogradely along the axon, deficits in axonal transport leading to a halt of axonal swellings have been described to precede AxD in experimental models of amyotrophic lateral sclerosis, multiple sclerosis, oxidative stress, and genetic models [39,40,41,42].

Our AxD time-lapse data revealed different morphological patterns of degeneration that can occur in the same axons over time (Figure 5 and Appendix A). We categorized these morphological patterns as:Granular degeneration: AxD instantly resulting in granular separated fragments.Retraction degeneration: AxD in which the distal part of the axon retracts, ultimately resulting in granular degeneration.Swelling degeneration: AxD in which axonal swellings enlarge, followed by granular degeneration.Transport degeneration: AxD in which the transport of axonal swellings of constant size, which do not enlarge, is halted, ultimately resulting in granular degeneration.

We trained a RNN, the EntireAxon RNN, to identify these morphological patterns based on changes in class segregation over time using a training dataset of AxD segmentation recordings (Figure 6a). This training dataset contained 162 time series of individual, degenerating axons and the corresponding segmentation masks generated by the EntireAxon CNN that were manually labeled according to the different degeneration patterns. Given the 4 different classes (background, axon, axonal swelling, and axonal fragment), 16 different class pairs can occur between a segmentation at time step *t* and time step *t* + 1. For example, a background pixel at *t* can either remain background pixel at *t* + 1 or change into one of the other three classes, and the same is true for the other classes. Thus, in total, 4 times 4 class pairs are possible. We used a window size of 32 × 32, of which the probability of a class pair in the central pixel relative to the previous timepoint was computed for each timepoint and across the entire image.

The RNN determined seven clusters (cluster 0–6) that were characterized by an idiosyncratic pattern of the changes in class distribution over 24 h (Appendix A). All clusters showed a decrease in the class ‘axon’ and an increase in the class ‘background’. Depending on the hemin concentration, the changes occurred at a different magnitude and at different timepoints, with concomitant increases in either the class ‘axonal swelling’ and/or ‘axonal fragment’. In cluster 0, there was an early decrease in the class ‘axon’, which then continued more linearly, as well as a later rise in the class ‘axonal fragment’. In contrast to cluster 0, cluster 1 showed no increase in the class ‘axonal fragment’ and a linear decrease in the class ‘axon’ from the start. In cluster 2, there was a strong increase in the class ‘axonal swelling’. Cluster 3 demonstrated an early and lasting high level of the class ‘axonal swelling’, with a later increase in the class ‘axonal fragment’. Cluster 4 showed a rapid decrease in the class ‘axon’ concomitant with increases in the classes ‘background’ and ‘axonal swelling’. Cluster 5 was similar to cluster 1, but with an early drop in the class ‘axon’. Cluster 6 showed an increase in the class ‘axonal swelling’ similar to but to a greater extent than cluster 2.

The RNN categorized each cluster to one of the four morphological patterns (Figure 6b): (i) Granular degeneration was defined by the clusters that described the degeneration of axons into axonal fragments, i.e., clusters 0, 1, 3, and 5. (ii) Retraction degeneration only included clusters 1 and 5, indicating the retraction of the axon followed by its fragmentation. (iii) Swelling degeneration was characterized by the three clusters that included the class ‘axonal swelling, i.e., clusters 2, 3, and 6, as well as cluster 5, showing the exchange of the class ‘axon’ for ‘background’. (iv) Transport degeneration was the only pattern that relied on cluster 4 and was also partly characterized by clusters 0, 1, 2, and 6. Although some clusters overlapped among morphological patterns, the unique combination of the different clusters allowed us to distinguish all four morphological patterns.

To validate the EntireAxon RNN, a 10-fold cross-validation was performed. Therefore, the dataset was randomly divided into 10 datasets and ten models were trained with 9 of the datasets leaving the remaining dataset for validation (not previously seen by the RNN). Based on the combined test samples, the RNN was able to distinguish between the four morphological patterns of AxD (Table 4). These data confirm that the combination of the different AxD features as well as their spatiotemporal progression defines distinct morphological AxD patterns.

### 3.4. The Morphological Patterns of AxD Depend on the Extent of AxD

We then applied the EntireAxon RNN to quantify the occurrence of the four morphological patterns of AxD in the context of hemorrhagic stroke (Figure 7 and Appendix A). Whereas all AxD patterns were detected (Figure 7a), hemin concentration dependently increased granular degeneration (*P* < 0.001), swelling degeneration (*P* < 0.001), and transport degeneration (*P* = 0.025, Figure 7b). When comparing the slopes of the different AxD patterns under hemin exposure, granular and swelling degeneration were significantly different from transport and retraction degeneration (*P* = 0.034 for granular vs. retraction degeneration, *P* = 0.026 for granular vs. transport degeneration, *P* = 0.030 for swelling vs. retraction degeneration, *P* = 0.026 for swelling vs. transport degeneration, Appendix A). Collectively, our data suggest that hemin concentration dependently induces different morphological patterns of AxD in cortical axons.

## 4. Discussion

We here describe two complementary tools, a novel microfluidic device and a deep learning algorithm, that allow increasing the experimental yield, in-depth enhanced throughput analysis of AxD, and longitudinal investigation of AxD in vitro. Using these tools, we were able to demonstrate time-dependent changes of the morphological features of AxD, with axonal swellings preceding axon fragmentation in a model of hemorrhagic stroke-induced AxD as well as the occurrence of four morphological patterns of AxD under pathophysiological conditions: granular, retraction, swelling, and transport degeneration.

We here propose a novel monolithic microfluidic device consisting of 16 individual microfluidic units that enables the parallel and separated treatment and/or manipulation of axons and somata (Figure 1). The currently available devices do not allow enhanced throughput experiments, as they comprise only single microfluidic units [19,23]. Although some devices can harbor multiple experimental conditions, they employ a radial design with a single soma compartment, in which one experimental condition may influence another due to the potential of retrograde signaling [22,43]. Another option is the parallel use of multiple individual devices, which allows handling up to 12 devices in a conventional 12-well plate [27]. Compared to our device, this procedure is time-consuming in both the manufacturing and adjustment for recordings.

To date, the extent of AxD has mainly been investigated with a focus on axon fragmentation as the primary readout. To quantify axon fragmentation, Sasaki and colleagues introduced the AxD index as the ratio of the fragmented axon area versus the total axonal area [26]. However, the AxD index did not include axonal swellings, which are a characteristic feature of degenerating axons [8,44]. Although other analyses have considered axonal swellings as a morphological feature of AxD [7,8], the approaches were time-consuming and required manual annotations.

We herein adapted a standard u-net with a ResNet-50 encoder [32,45] and used a CNN ensemble, which combines predictions from multiple CNNs to generate a final output and is superior to individual CNNs [46,47,48]. The EntireAxon CNN performs an automatic segmentation and quantification of axons and morphological features relevant to AxD, including axonal swellings and fragments, on phase-contrast time-lapse microscopy images (Figure 2). The EntireAxon CNN recognized the four classes—‘background’, ‘axon’, ‘axonal swelling’, and ‘axonal fragment’—with the highest mean F1 score for the class ‘background’ (Table 1). The comparably lower performance of the CNN to recognize axonal fragments may be explained by the disproportional distribution of pixels in the training and validation data (‘background’ mean of 96.42% of pixels, ‘axon’ 2.77%, ‘axonal swelling’ 0.58%, ‘axonal fragment’ 0.23%). Hence, every individual segmentation error affects the false positive or false negative rate more strongly in these classes.

The comparison with human evaluators revealed that the EntireAxon CNN reached a similar performance level. As expected, its performance was slightly better than the human evaluators on the ground truth, as both the ground truth and training data were labeled by the same human evaluator (Table 2). Interestingly, when comparing the EntireAxon CNN with a human evaluator on the consensus label of the other two human evaluators, not only was the EntireAxon CNN as good as or even better than the human evaluator, but the mean F1 scores were also higher than on the ground truth labels (Figure 3 and Table 3). This may be because pixels that were differentially assigned by the human evaluator, i.e., more difficult to classify, were excluded from the comparison. Taken together, these findings demonstrate that the EntireAxon CNN is suitable to automatically quantify AxD and its accompanying morphological changes in an enhanced throughput manner.

We then applied these novel tools to a model of hemorrhagic stroke-induced AxD by exposing axons from primary cortical neurons to the hemolysis product hemin and investigated the progression of AxD. Similar to previous results, where 100 µM hemin were sufficient to induce significant neuronal cell death in conventional cultures of somata and axons [15], we observed that 100 µM hemin led to a significant decrease in the axon area and an increase in the axonal swelling and fragment area (Figure 4).

It has been described previously that the progression of AxD undergoes a latent phase, during which the structural integrity of the axon is maintained, followed by a catastrophic phase with the rapid disintegration of the axon [8]. In our model, the catastrophic phase of AxD started between 12 and 18 h after the administration of hemin (Figure 4 and Appendix A). Similar durations of the latent phases of AxD have been observed in other models. For instance, under circumstances of growth factor withdrawal, the transition to the catastrophic phase occurred at 12–24 h [8,49,50]. Our findings are also in line with results reported in a model of experimental autoimmune encephalomyelitis, indicating that axonal swelling anticipates fragmentation [7]. Taken together, AxD progression depends on the severity of the insult, and axonal swellings may be reliable predictors of AxD.

Whereas we applied the EntireAxon CNN on time-lapse images, it should be noted that it is also possible to calculate the ratios between swelling or fragment area and axon area based on random images that are not necessarily in the same position over time. The respective areas are calculated for each image and class separately and independently of time or any previous images. However, we recommend including baseline recordings to allow for normalization due to differences in the neurite outgrowth in the devices.

Interestingly, axonal swellings and axonal fragments were related to different morphological patterns of AxD. Specifically, we observed axons that showed signs of axonal retraction, an enlarging of axonal swellings, and axonal transport before degeneration (Figure 5). We therefore trained the EntireAxon RNN to quantify the occurrence of four morphological patterns of AxD, i.e., granular, retraction, swelling, and transport degeneration, based on the clusters of unique changes of classes over time (Figure 6 and Appendix A). The RNN generated 7 clusters from the 16 possible class pairs. Of note, all clusters differed in the extent and timing of class changes. The RNN used a different ensemble of clusters to classify each pattern. However, some clusters overlapped between patterns, which can be explained by the fact that the eventual degeneration of the axon, which leads to fragments, is the end-product of AxD. Therefore, clusters 0, 1, 3, and 5 also occur in retraction, swelling, and transport degeneration. On the other hand, axonal swellings are a dominant feature of swelling and transport degeneration, which explains why cluster 6 was used to define both patterns.

These different AxD patterns have not yet been described to occur simultaneously in the same biological condition. However, granular degeneration was observed in retrograde, anterograde, Wallerian, and local AxD after axotomy or trophic factor deprivation [6,10,12,13]. Retraction degeneration was described in axonal retraction and shedding in developmental AxD [9,11]. Swelling degeneration was reported in experimental autoimmune encephalitis and growth factor deprivation [7,8]. Transport degeneration was shown in experimental models of amyotrophic lateral sclerosis, multiple sclerosis, oxidative stress, and genetic models [39,40,41,42].

Our data demonstrate that all four morphological degeneration patterns can occur along cortical axons (Figure 7). Interestingly, we also observed a concentration-dependent effect in the context of hemorrhagic stroke. Granular, swelling, and transport degeneration were significantly increased with increasing hemin concentrations, with granular and swelling degeneration being more strongly correlated. The extent to which our model of hemin-induced AxD in hemorrhagic stroke is molecularly similar to developmental or pathophysiological AxD needs to be further investigated, along with the underlying molecular mechanisms of the four patterns of AxD. This could be greatly facilitated by the EntireAxon RNN, which is able to automatically detect the morphological patterns in time-lapse recording due to its capacity to relate each output to previous images in the stacks by its current units.

### Limitations and Outlook

Our microfluidic device does not currently allow the investigation of AxD at more proximal axonal parts to the soma, such as the axonal initial segment. Shortening the length of the microgrooves or including a more proximal compartment are possible modifications of the current design.Our results are based on unmyelinated axons. Co-culture with glia cells that may play a role in AxD is possible in the presented microfluidic device, and the time course and morphological changes may be different under different co-culture conditions. These studies are of high relevance to the field but are beyond the scope of the present study.The observed effects of AxD in hemorrhagic stroke within this study were based on hemin toxicity, and we cannot exclude that other hemolysis products, such as thrombin or bilirubin, have different effects. Additional studies should investigate differences of hemolysis products to increase our understanding of the mechanisms of AxD in hemorrhagic stroke.The overall CNN performance may be further improved with more general inputs. For example, the segmentation of fragment pixels cannot be accurately conducted based on a single image at a specific timepoint. Instead, the whole process of AxD, ultimately resulting in the disintegration of the axons (i.e., the generation of axonal fragments), needs to be considered. In principle, CNNs using 3D convolutions could perform a segmentation over an entire time-lapse recording and model the temporal dependencies. However, we decided against the 3D approach, as it severely restricts general applicability due to its greatly increased effort to label suitable time series for training. In this context, the identification of the images that will yield the best results is crucial to effectively reduce labeling costs, which we have described previously using an active learning method [51].

## 5. Conclusions

The EntireAxon platform, the combination of an advanced microfluidic device and a deep learning tool, expands our possibilities to track AxD by detecting axons, axonal swellings, and axonal fragments in an enhanced throughput manner. We further identified four morphological patterns of AxD, i.e., granular, retraction, swelling, and transport degeneration, under pathophysiological conditions in the context of hemorrhagic stroke. The EntireAxon platform will help to tackle the complex processes of AxD and may significantly enhance our understanding of AxD in health and disease to develop novel therapeutic strategies for brain diseases.

## 6. Patents

A.P. (Alex Palumbo), P.G., and M.Z. declare that they have filed a patent for the microfluidic device and the EntireAxon deep learning algorithm to quantify axonal degeneration (European Patent Office, file number: 20152016.0).

## Figures and Tables

**Figure 1 cells-10-02539-f001:**
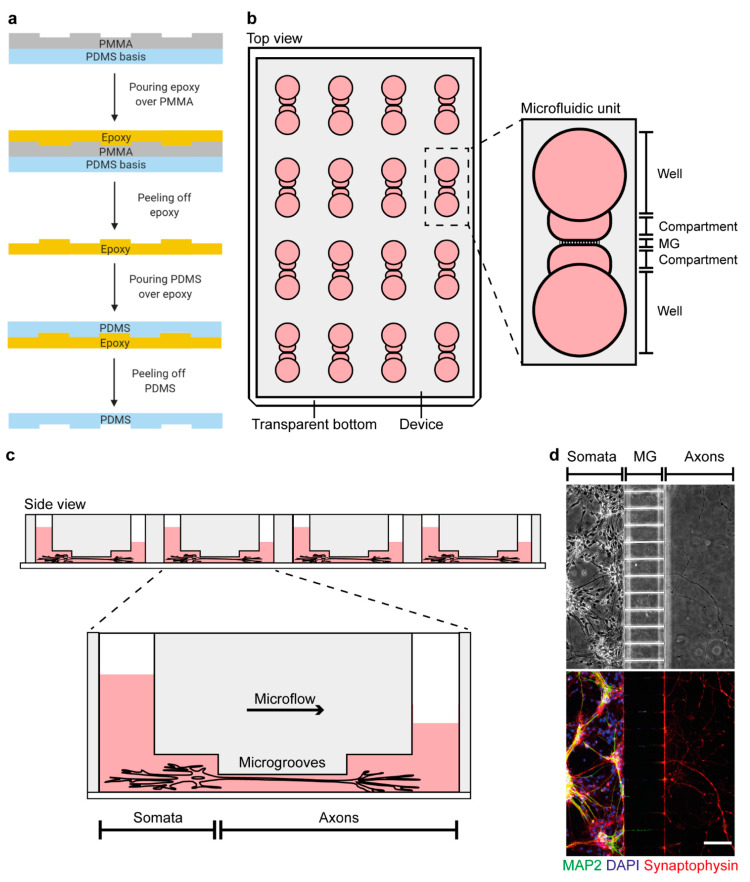
Microfluidic device for the enhanced throughput cultivation of axons. (**a**) Manufacturing process of the microfluidic device prototype created with BioRender.com. (**b**) The microfluidic device incorporates 16 individual microfluidic units for axon cultivation. One microfluidic unit consists of two wells that are connected through compartments and microgrooves (MG). (**c**) Primary cortical neurons are seeded into the soma compartment from which their axons grow through the MG into the axon compartment. Directed growth is supported by culture medium microflux due to different medium volumes between the two wells. (**d**) Phase-contrast image of primary cortical axons that were spatially separated from their somata by the MG at day in vitro 7, which we confirmed by immunofluorescence staining of dendrites using microtubule-associated protein 2 (MAP2, green, 1:4000) and axons using synaptophysin (red, 1:250). DAPI (blue, 1:1000) was used for nuclear counterstaining (top). Scale bar: 100 µm.

**Figure 2 cells-10-02539-f002:**
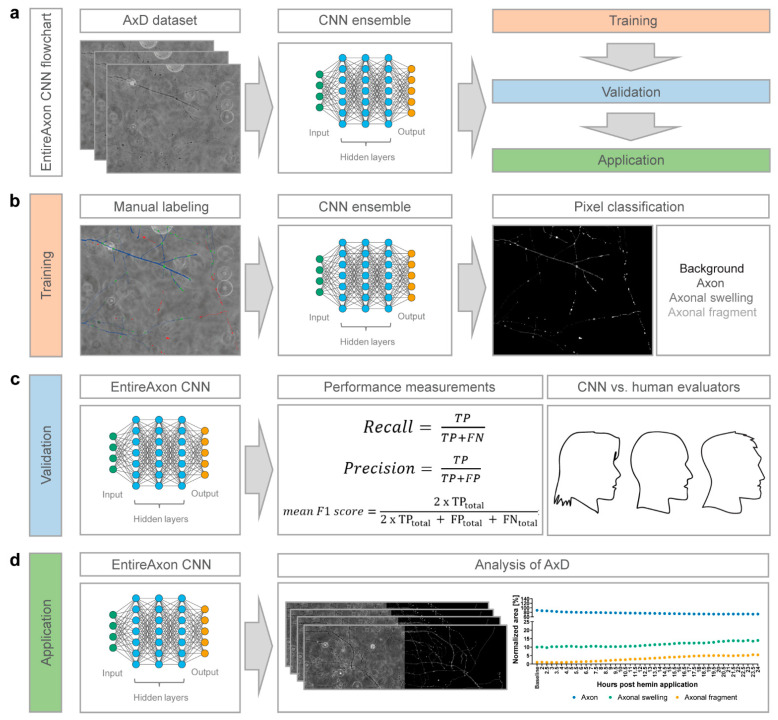
EntireAxon CNN for the enhanced throughput analysis of AxD. (**a**) The flowchart of the EntireAxon CNN. The AxD data was separated into training, validation, and testing data. We adapted a standard u-net with a ResNet-50 encoder and used a CNN ensemble, which combines the predictions from multiple CNNs to generate a final output and is superior to individual CNNs. (**b**) We manually labeled the training data to segment each pixel into the four classes ‘background’, ‘axon’, ‘axonal swelling’, and ‘axonal fragment’, which are displayed in the output image in black, dark grey, intermediate grey, and light grey, respectively. We trained an ensemble comprising eight CNNs to segment the four classes. (**c**) The EntireAxon CNN was validated with a separate validation dataset to assess its performance (recall, precision, and mean F1 score), which was compared to the human evaluators (ground truth was labeled by human evaluator 1). (**d**) The EntireAxon CNN was applied to data on AxD induced by the exposure of hemin, which is used to model of hemorrhagic stroke in vitro.

**Figure 3 cells-10-02539-f003:**
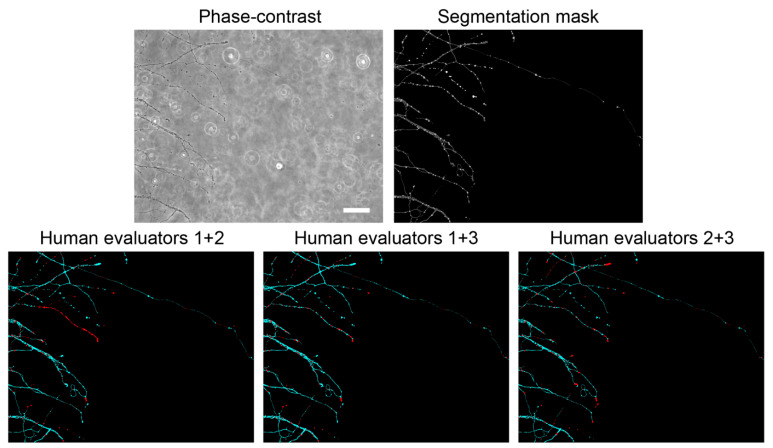
Performance of the EntireAxon CNN compared to human evaluators. Phase-contrast validation image, its EntireAxon CNN segmentation mask, and the consensus labeling masks of two human evaluators that show the segmentation overlap (cyan) or difference (red) between the labels. Scale bar: 100 µm.

**Figure 4 cells-10-02539-f004:**
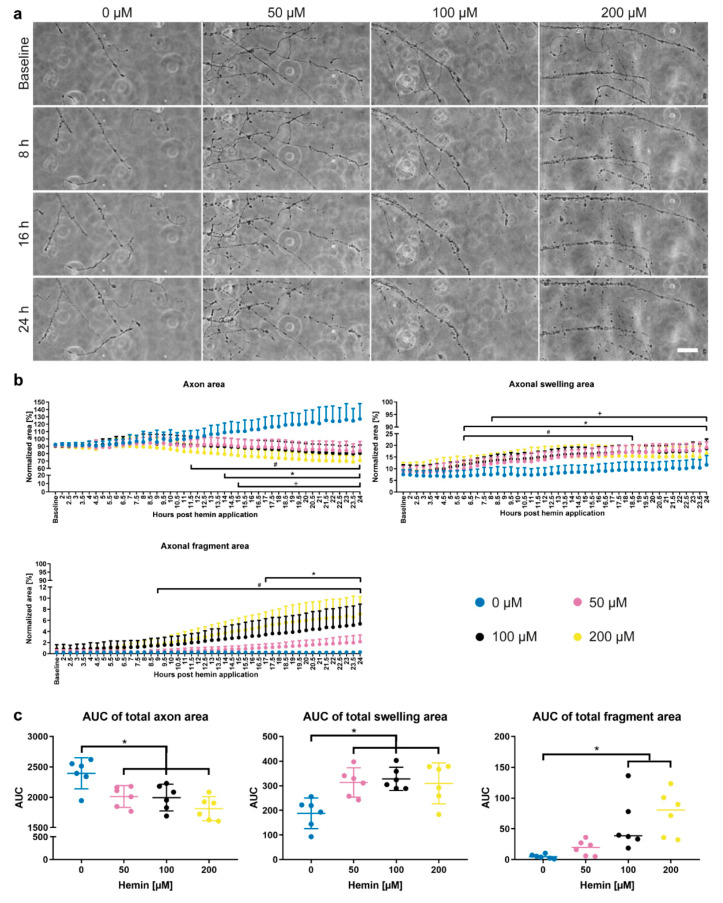
Time-dependent hemin-induced AxD. (**a**) Primary cortical axons treated with hemin (50, 100, 200 µM) degenerated compared to vehicle-treated axons (0 µM) that continued to grow. Scale bar: 50 µm. For complete time-lapse videos including segmentation, refer to Appendix A. (**b**) Quantification of AxD over 24 h in phase-contrast images. To determine the time course, the sum of pixels in each class and hemin concentration over time was normalized to the baseline of that class and condition. The quantification of the phase-contrast images over 24 h revealed significantly smaller axon areas starting at 11.5 h after 200 µM (*P* = 0.020), at 14 h after 100 µM (*P* = 0.040), and at 15 h after 50 µM (*P* = 0.018) hemin treatment compared to the control (0 µM). The axonal fragment area significantly increased from 9.5 h onwards in 200 µM hemin (*P* = 0.037) and from 17.5 h in 100 µM hemin (*P* = 0.044), while the axonal swelling area increased from 6 h onward in 100 µM hemin (*P* = 0.019) and 200 µM hemin (*P* = 0.010), and from 8 h in 50 µM hemin (*P* = 0.030). *N* = 6 independent cultures of primary cortical neurons. Means + 95% CI are given. One-way ANOVA with Greenhouse-Geisser correction. +, *, # *P* < 0.05; + = 50 µM vs. 0 µM, * = 100 µM vs. 0 µM, # = 200 µM vs. 0 µM. For detailed statistical information, refer to Appendix A. (**c**) Area-under-the-curve (AUC) analysis of hemin-induced AxD. Whereas axons exposed to hemin showed a decline in axon area, axonal swelling and axonal fragment area increased over 24 h. *N* = 6 independent cultures of primary cortical neurons. Means ± 95% CI are given for axon and axonal swelling area, medians for fragment area. * *P* < 0.05 vs. 0 µM. For exact *P* values, refer to Appendix A.

**Figure 5 cells-10-02539-f005:**
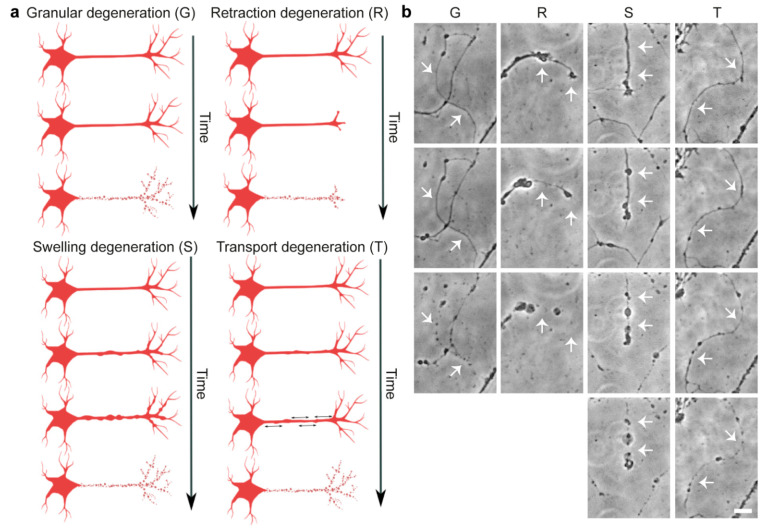
Four morphological patterns of AxD. (**a**) Schematic overview and (**b**) phase-contrast recordings of the proposed AxD morphological patterns in primary cortical axons: Granular degeneration (G) is characterized by the fragmentation of the axon (white arrows). During retraction degeneration (R), the axonal growth cone retracts in the proximal direction, and the part of the axon in proximity of the growth cone disintegrates along with the axonal swellings (white arrows). During swelling degeneration (S), many axonal swellings enlarge, resulting in axonal fragments (white arrows). During transport degeneration (T), the transport of axonal swellings along the axon is halted prior to the degeneration of the axon (white arrows). Scale bar: 20 µm. For complete time-lapse videos including segmentation, refer to Appendix A.

**Figure 6 cells-10-02539-f006:**
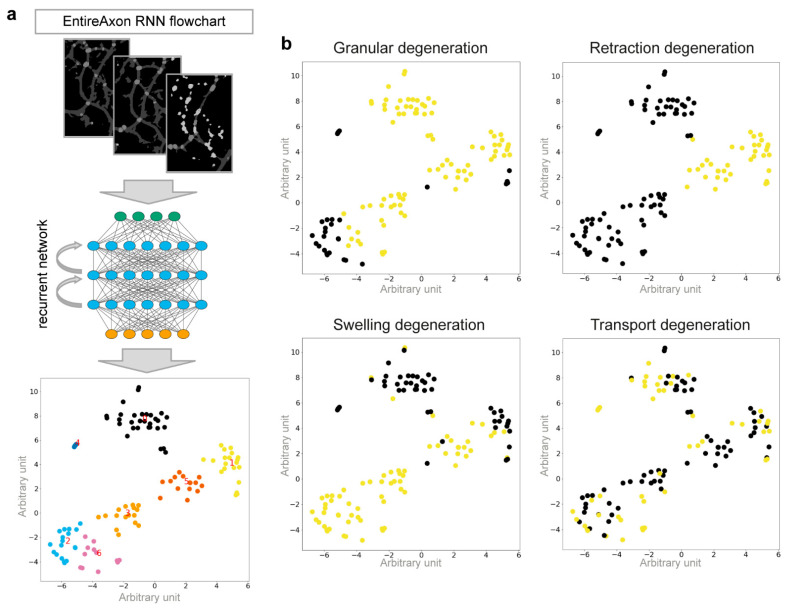
Recognition of four morphological patterns of AxD by the EntireAxon RNN. (**a**) Schematic workflow of the RNN to recognize and quantify morphological patterns of AxD based on the identification of seven clusters. The EntireAxon CNN segmentation masks were used for the RNN training, which determined the change in class over time. Based on the 16 different possible class pairs, the RNN determined 7 clusters (cluster 0–6). To visualize the relationships of the specific samples, we employed t-distributed stochastic neighborhood embedding (T-SNE) to compute a two-dimensional representation of the high-dimensional data. (**b**) The clusters classify the four morphological patterns of AxD with yellow indicating included and black indicating excluded clusters: granular (G), retraction (R), swelling (S), and transport degeneration (T). Clusters of granular degeneration overlap with recognized clusters of other morphological patterns (retraction, swelling, and transport degeneration). For more details on the morphological changes underlying the cluster analysis, refer to Appendix A.

**Figure 7 cells-10-02539-f007:**
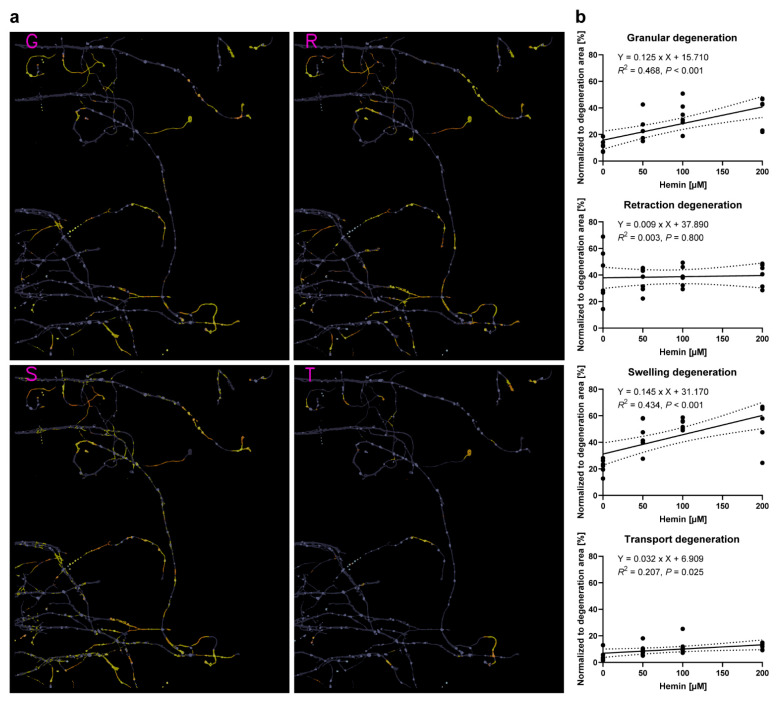
Concentration-dependent differences in the morphological patterns of hemin-induced AxD. (**a**) The classification of granular (G), retraction (R), swelling (S), and transport degeneration (T) in primary cortical axons treated with 200 µM hemin. For the complete time-lapse video including segmentation, refer to Appendix A. (**b**) Linear regressions of the four morphological patterns of AxD in hemin-induced AxD. The area classified for each AxD pattern was normalized to the total degeneration area. Dotted lines show 95% confidence bands. *N* = 6 independent cultures of primary cortical neurons. Granular degeneration: F(1, 22) = 19.330, *P* < 0.001. Retraction degeneration: F(1, 22) = 0.066, *P* = 0.800. Swelling degeneration: F(1, 22) = 16.900, *P* < 0.001. Transport degeneration: F(1, 22) = 5.757, *P* = 0.025. When comparing the slopes of the different AxD patterns, granular and swelling degeneration were significantly different from transport and retraction degeneration (*P* = 0.034 for granular vs. retraction degeneration, *P* = 0.026 for granular vs. transport degeneration, *P* = 0.030 for swelling vs. retraction degeneration, *P* = 0.026 for swelling vs. transport degeneration). For detailed statistical analysis, refer to Appendix A.

**Table 1 cells-10-02539-t001:** Validation of the EntireAxon CNN performance for all four classes—‘background’, ‘axon’, ‘axonal swelling’, and ‘axonal fragment’—in previously unseen phase-contrast microscopic images.

Class	Precision	Recall	Mean F1-Score
Background	0.993	0.996	0.995
Axon	0.789	0.774	0.780
Axonal swelling	0.609	0.534	0.567
Axonal fragment	0.805	0.196	0.301

**Table 2 cells-10-02539-t002:** Comparison of the mean F1 scores between the EntireAxon CNN and two human evaluators on the ground truth (human evaluator 1 also labeled the training images) to recognize background, axon, axonal swelling, and axonal fragments.

	Mean F1-Score			
Class	Background	Axon	Axonal Swelling	Axonal Fragment
EntireAxon CNN	0.995	0.780	0.567	0.301
Human evaluator 2	0.991	0.654	0.485	0.548
Human evaluator 3	0.993	0.704	0.489	0.221

**Table 3 cells-10-02539-t003:** Comparison of the mean F1 scores between the EntireAxon CNN and the human evaluator on the consensus labeling of the other two human evaluators.

		Mean F1-Score			
Consensus	Class	Background	Axon	Axonal Swelling	Axonal Fragment
Evaluators 1 and 2	EntireAxon CNN	0.998	0.847	0.667	0.400
	Evaluator 3	0.998	0.808	0.647	0.376
Evaluators 1 and 3	EntireAxon CNN	0.998	0.870	0.710	0.674
	Evaluator 2	0.996	0.759	0.716	0.564
Evaluators 2 and 3	EntireAxon CNN	0.998	0.781	0.607	0.590
	Evaluator 1	0.996	0.747	0.592	0.421

**Table 4 cells-10-02539-t004:** Ten-fold cross-validation of the four morphological patterns of AxD.

Class	Precision	Recall	Mean F1-Score
Granular degeneration	0.796	0.953	0.868
Retraction degeneration	0.419	0.500	0.456
Swelling degeneration	0.681	0.681	0.681
Transport degeneration	0.442	0.824	0.575

## Data Availability

All data needed to evaluate the conclusions in the paper are present in the paper and/or the Appendix A. The time-lapse data and code are available upon reasonable request to the corresponding authors. We plan to launch a website to enable other researchers to use the tool.

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
