# Peer review of "Deep Learning to Decipher the Progression and Morphology of Axonal Degeneration"

_cells, 2021, doi:10.3390/cells10102539_

Round 1

Reviewer 1 Report

Here, Palumbo and colleagues present a novel approach to quantify axon degeneration on phase contrast microscopy images. They apply this to assess axon degeneration in cortical neurons in an in vitro model of haemorrhagic stroke induced by hemin. They also use a custom-built microfluidic device which allows to run multiple cultures and experiments in parallel.

The topic of the research is important; current methods to analyse axon degeneration can be used to reliably detecting degenerated axons, but are not particularly good to detect milder changes in unhealthy axons that are not degenerated yet or will not progress to clear fragmentation.

In principle, the novel approach the authors propose could help fill this gap, providing more detailed information on the “health status” of the axons. However, as it stands, I feel it could only be applied in limited circumstances. A couple of suggestions that could, in my opinion, really strengthen the study.

Major points:

-As presented, the approach seems to require time-lapse imaging of the same field over time, as everything is normalised to baseline. Although imaging the same field over time is the most accurate way to assess degeneration, this is not always possible (at least not exactly the same field) and many laboratories working on axon degeneration would not use this approach. Can the authors think of a way to extend this approach to quantify axon degeneration in images from random field of axons taken hours apart? For instance, ratio between swelling area/axon area could be important.

-Related to the comment above, other types of primary neurons will grow axons much faster than cortical. Would this approach work also in these circumstances? Over longer timeframes, the fields imaged would look quite different from baseline due to many more axons crossing the field. Although not strictly necessary, if the authors have access to images from other primary neuron types, it would be really interesting to try this approach with other types of neurons.

-I am unsure about the different patterns of degeneration detected. It is quite difficult to distinguish granular degeneration from swelling degeneration already; are they actually 2 different processes? In addition, I strongly recommend removing transport degeneration. These “swellings” moving around the axon can be seen in vitro in healthy cultures too; it is hard to link them to a degeneration pattern in my opinion.

Minor suggestion:

-Would be beneficial to merge figure 4 and 5 since they are part of the same analysis, maybe by reducing the number of phase contrast images shown in main figure.

Strengths of the study:

-I really appreciate that the authors focus on phase contrast images. I believe this is the best and more convenient way to assess axon degeneration, as no fixation of the sample is needed (as the author state).  

-The fact that their approach works on images of axons grown in microfluidic devices is promising. In these kinds of experiments, the quality of the images is normally lower due to the presence of the microfluidic chamber. This means that the quantification is likely to be even more accurate for experiments on neurons cultured in a normal dish.

Reviewer 2 Report

In this paper, Palumbo et. al. explored the features of axonal degeneration and use of machine learning to identify the features of axonal degeneration. The authors used a novel microfluidic device to do the experiments of axonal degeneration, and analyzed the microscopic images with a trained convolutional neural network (CNN). They found that their system can identify the axonal damage, rather with a better efficiency than human evaluators. This is an important paper that can be very useful for the pathological evaluation of not only neurodegenerative disease, but also other diseases of the brain.

The design of the study is sound, data shown here is of high quality. The only problem I found is that sometimes the authors added some sentences that should be discussed in the introduction or ‘discussion’ section.

Example:

‘Several cell culture systems such as Campenot chambers and microfluidic devices 410 have been developed to spatially separate axons from their somata allowing to study AxD 411 at a molecular level and to dissect the axon-soma relationship in AxD [32,33]. While 412 Campenot chambers facilitate the collection of axonal material due to their open structure 413 [34], they are only suitable for neurons that project axons robust enough to cross the vac-414 uum grease barrier that is needed to affix the Teflon piece that separates the somata from 415 their axons. This makes Campenot chambers unsuitable for most central nervous system 416 neurons [35]. In microfluidic devices, the spatial separation is enabled by a microflux es-417 tablished by a medium volume difference between the two opposing compartments that 418 are separated by microgrooves through which axons grow [36]. 419

The major limiting factor of commercially available microfluidic devices to study 420 AxD is that they are single, individual systems and hence, can only be used to assess one 421 condition, which is time-consuming and precludes high-throughput analyses [33,37]’

These types of descriptions should be moved to the ‘introduction’ or ‘discussion’ section.

Also, I could not see any of the videos provided by the authors as supplemental materials

I found this paper interesting. It should be accepted after minor revision.
